# Peptides: Prospects for Use in the Treatment of COVID-19

**DOI:** 10.3390/molecules25194389

**Published:** 2020-09-24

**Authors:** Vladimir Khavinson, Natalia Linkova, Anastasiia Dyatlova, Boris Kuznik, Roman Umnov

**Affiliations:** 1Department of Biogerontology, Saint Petersburg Institute of Bioregulation and Gerontology, 197110 Saint Petersburg, Russia; vladimir@khavinson.ru (V.K.); nasya-nastasya@yandex.ru (A.D.); t.kmb@mail.ru (R.U.); 2The Group of Peptide Regulation of Aging, Pavlov Institute of Physiology of RAS, 199034 St. Petersburg, Russia; 3Department of Therapy, Geriatry, and Anti-Aging Medicine, Academy of Postgraduate Education under FSBU FSCC of FMBA of Russia, 125310 Moscow, Russia; 4Department of Medical and Biological Disciplines, Belgorod State University, 308015 Belgorod, Russia; 5Department of the normal physiology, Chita State Medical Academy, 672000 Chita, Russia; bi_kuznik@mail.ru

**Keywords:** COVID-19, immunity, hemostasis, drugs, peptides, immunomodulators

## Abstract

There is a vast practice of using antimalarial drugs, RAS inhibitors, serine protease inhibitors, inhibitors of the RNA-dependent RNA polymerase of the virus and immunosuppressants for the treatment of the severe form of COVID-19, which often occurs in patients with chronic diseases and older persons. Currently, the clinical efficacy of these drugs for COVID-19 has not been proven yet. Side effects of antimalarial drugs can worsen the condition of patients and increase the likelihood of death. Peptides, given their physiological mechanism of action, have virtually no side effects. Many of them are geroprotectors and can be used in patients with chronic diseases. Peptides may be able to prevent the development of the pathological process during COVID-19 by inhibiting SARS-CoV-2 virus proteins, thereby having immuno- and bronchoprotective effects on lung cells, and normalizing the state of the hemostasis system. Immunomodulators (RKDVY, EW, KE, AEDG), possessing a physiological mechanism of action at low concentrations, appear to be the most promising group among the peptides. They normalize the cytokines’ synthesis and have an anti-inflammatory effect, thereby preventing the development of disseminated intravascular coagulation, acute respiratory distress syndrome and multiple organ failure.

## 1. Introduction

COVID-19 is an acute respiratory disease caused by a SARS-CoV-2 virus (severe acute respiratory syndrome coronavirus 2), the outbreak of which began in late December 2019 in China; in Spring 2020 it was declared a pandemic [1,2]. SARS-CoV-2 is an RNA-containing virus from the coronavirus family, which includes viruses causing mild respiratory symptoms, pathogens of SARS (2002–2003 epidemic) and the Middle East respiratory syndrome MERS-CoV (2005 epidemic).

Based on phylogenetic analysis, coronaviruses include four groups: alphacoronavirus (αCoV), betacoronavirus (βCoV), gammacoronavirus (γCoV) and deltacoronavirus (δCoV). Coronaviruses αCoV and βCoV affect only mammals, and γCoV and δCoV affect birds; however, some of them can also be contracted by mammals. α- and β-coronaviruses cause respiratory diseases in humans and gastroenteritis in animals [3].

The genome of the virus isolated from a group of patients with SARS is 82% identical to the human virus SARS-CoV [4]. For this reason, the virus was called SARS-CoV-2 [5]. The SARS-CoV-2 genome is 96.2% identical to the genome of the RaTG13 virus of bats Rhinolophus affinis [6]. It is still unknown which animals can be intermediate hosts of SARS-CoV-2. In the analysis of 103 SARS-CoV-2 genomes, two strains were isolated: type L and type S. Strain L mutated from S [7]. Which of the SARS-CoV-2 coronavirus strains is more pathogenic is currently unknown.

The main transmission route of SARS-CoV-2 is considered to be airborne; contact and fecal-oral methods of transmission are also possible [8,9]. The latent period of the disease lasts from 2 to 14 days. The propagation rate of SARS-CoV-2 is several times higher than that of SARS-CoV. In approximately 44% of cases, coronavirus infection is transmitted from people in the latent period, without any clinical signs of infection [10].

As of the beginning of September 2020, the COVID-19 pandemic has affected more than 27 million people in 208 countries. More than 880 thousand people have died; more than 18 million have people recovered. In Russia, the number of cases has reached more than 1 million people. Despite the efforts of health workers, the death toll in the world is increasing every day. The treatment of COVID-19 remains symptomatic so far. Currently there is no clinically approved antiviral drug or vaccine against COVID-19. The review discusses the molecular aspects of the SARS-CoV-2 action and the prospects for studying peptides as potential drugs for the complex therapy of COVID-19.

## 2. Main Text

### 2.1. Prospective Molecular Mechanisms for the Pathogenesis of Coronavirus Infection Caused by SARS-CoV-2

The structure, life cycle and molecular targets of SARS-CoV-2 in host cells are shown in the diagram (Figure 1). Each stage of the SARS-CoV-2 coronavirus life cycle represents a potential target for drug therapy. The penetration of the virus into the host cell membrane occurs in two stages: the first stage involves contacting the receptor on the surface of the target cell; the second stage is the fusion of the virus with the cell membrane, either on the cell surface or inside of it. In case of SARS-CoV-2, the first step requires that the spike proteins undergo biochemical modification, the so-called priming of protein S [11,12,13].

The viral spike SARS-CoV utilizes angiotensin-converting enzyme 2 (ACE2) as an input receptor [14] and uses cell serine proteases TMPRSS2 and cathepsin L for priming [15,16,17]. The functions of the TMPRSS2 protein remain insufficiently studied. Overexpression of the TMPRSS2 gene under the action of androgens in prostate cancer cells and reduced expression of the TMPRSS2 gene in androgen-independent prostate cancer were detected [18]. In the case of SARS-CoV-2, TMPRSS2 is responsible for the priming of the viral protein spike, which entails cleavage of the spike at two sites: Arg685/Ser686 and Arg815/Ser816.

The TMPRSS2 and virus spike binding model was created. It was found that TMPRSS2 interacts with two spike loops. The key functional residues of TMPRSS2 (His296, Ser441 and Ser460) interact with the flanking residues of the spike cleavage sites. It is assumed that the TMPRSS2 region, which interacts with the C-terminal cleavage site (Arg815/Ser816) of the spike, is relatively more suitable for creating targeted drugs compared to the TMPRSS2 region, which interacts with the *N*-terminal cleavage site (Arg685/Ser686) [19].

The role of cathepsin L in the SARS-CoV-2 spike priming is suggested based on the data on its participation in the SARS-CoV priming [20,21,22]. At high (alkaline) pH, the fusion of the viral membrane and the target cell does not occur. Thus, protonation can indirectly facilitate the penetration of SARS-CoV into the cell. Inhibitors of cathepsin L activity block the introduction of SARS-CoV into target cells [22]. Probably, protonation is necessary for the activation of cathepsin L, rather than a spike protein. These results showed that cathepsin L is a SARS-CoV-activating protease.

The efficiency of ACE2 receptor recruitment is a key factor in determining the ability of SARS-CoV to penetrate the cell [23]. Viral spikes SARS-CoV and SARS-CoV-2 have an amino acid identity of 72%; therefore, it is assumed that the mechanisms of penetration of SARS-CoV and SARS-CoV-2 are similar. However, there is a characteristic loop with flexible glycine residues in the SARS-CoV-2 structure, as opposed to hard proline residues in the same SARS-CoV region. Molecular modeling revealed a stronger interaction of RBD SARS-CoV-2 with the ACE2 receptor. Phenylalanine F486, located in a flexible loop, probably plays a major role in this interaction, since it penetrates the hydrophobic “pocket” of ACE2 [24]. The SARS-CoV-2 viral spike has higher affinity for the human ACE2 receptor compared to the Bat-CoV viral spike [25,26].

ACE2′s active extracellular domain is exposed on the surfaces of the alveolar epithelial cells of the lungs, and on the endothelial cells of arteries and veins, smooth muscle arterial cells, renal tubule epithelium and small intestine epithelium [27,28]. In addition, ACE2 gene expression was found in the epithelium of the upper respiratory tract, central nervous system (CNS) organs, pigmented epithelial cells of the eyes and liver hepatocytes [17].

In patients with COVID-19, an increase in the level of angiotensin II (Ang II) in the blood was detected, which indicates a suppression of the ACE2 expression in the tissues. This leads to dysfunction of the renin-angiotensin system (RAS). It should be noted that RAS is a hormonal system in humans and mammals that regulates blood pressure and blood volume in the body through a peptide cascade. Several key molecules are distinguished in RAS: renin, angiotensin I and its fragment Ang I (1–10), angiotensin-converting enzyme (ACE), angiotensin II. Renin is released from the secretory granules of juxtaglomerular cells in the event of a decrease in perfusion pressure or NaCl concentration. ACE hydrolyses Ang I (1–10) to angiotensin II, a polypeptide with potent vasoconstricting properties. It should be noted that the activation of RAS raises blood pressure.

ACE2 is an ACE homolog and a negative regulator of RAS. The function of ACE2 is the degradation of angiotensin II, which is a vasoconstrictor and has pro-fibrous and pro-inflammatory properties, to Ang (1–7), a vasodilator with antiproliferative and apoptotic properties. In addition to the systemic function in maintaining blood pressure, ACE2 has local regulatory effects in pathological changes of internal organs [29].

ACE2-expressing organs are targets for SARS-CoV-2. Lungs serve as the primary target of the virus, as evidenced by the severity of respiratory symptoms and the development of pneumonia in severe COVID-19 cases. Pathomorphological studies revealed diffuse alveolar lesions with cell fibromyxoid exudate in the lungs of patients with COVID-19 [1,11,30,31]. Additionally, 83% of ACE2 RNA is expressed on the apical surface of type II human lung alveolar cells (AT2). ACE2 RNA is also expressed on other cell types: type I alveolar cells (ATI), airway epithelial cells, fibroblasts, endotheliocytes and macrophages—but in smaller amounts in comparison with the AT2 [32].

There is evidence of secondary symptoms of COVID-19, associated with multi-organ expression of ACE2. Some patients have acute cardiac and renal failure [31,33]. This may be due to the fact that 4 to 7.5% ACE2 mRNA was found in cardiomyocytes and in the proximal renal tubule cells. ACE2 expression was also found in the villi of the placenta chorion and uterus [34,35], making the transfer of COVID-19 from mother to fetus possible.

Thus, the effect of SARS-CoV-2 on ACE2 and subsequent disruption of RAS regulation contributes to the development of a multiple organ damage [36]. In this regard, patients with diabetes mellitus, hypertension and lung diseases are exposed to the highest risk of COVID-19 [33,37]. A favorable condition for the penetration of SARS-CoV-2 into a cell is a low cytosolic pH, which is typical for diabetes mellitus, hypertension and obesity. In people of older age groups and for those suffering from nicotine addiction, the cytosolic pH is also lowered [38].

A potential target for pharmacotherapy with COVID-19 may be the CD147 molecule (basigin, EMMPRIN, extracellular matrix metalloproteinase inducer). CD147 is a highly glycosylated transmembrane protein of the immunoglobulin superfamily, which acts as an activator of matrix metalloproteinases (MMPs) and can increase their expression in asthmatic and diabetic complications. The expression levels of CD147 and MMPs are often increased in inflammatory processes and are associated with cancer progression [39]. In patients with severe asthma, a high level of MMP-9 in sputum was detected [40]. Influenza A virus infection increases the expression of CD147 in lung cells in asthmatics [41].

Blocking of CD147 with antibodies (meplazumab) leads to a dose-dependent inhibition of SARS-CoV-2 replication. Co-immunoprecipitation, ELISA and immuno-electron microscopy revealed the interaction between SARS-CoV-2 spike and CD147 [42]. Based on these data, it is assumed that meplazumab is a potential drug for the treatment of COVID-19. Clinical trials of meplazumab are currently underway in China to treat patients infected with SARS-CoV-2 (ClinicalTrials.gov ID: NCT04275245).

It was shown that lethality from SARS-CoV-2 is due to acute respiratory distress syndrome caused by the cytokine release syndrome (CRS). According to this hypothesis, the virus enters a cell through ACE2 and TMPRSS2, whereat the virus RNA begins to act as a pathogen-associated molecular pattern (PAMP) and is recognized by toll-like receptors (TLRs). It is suggested that rapid virus replication can cause apoptosis of epithelial and endothelial cells, leading to the secretion of pro-inflammatory cytokines and chemokines. SARS-CoV-2 can also stimulate pyroptosis of macrophages and lymphocytes [43,44]. This leads to a release of chemokines, which causes the migration and activation of neutrophils. In support of the hypothesis on the CRS role in the pathogenesis of COVID-19, it was shown that patients with severe forms of the disease manifested higher levels of interleukins (IL-6, IL-10, IL-2 and IFN-γ), GCSF, IP-10, MCP-1, MIP-1A and tumor necrosis factor (TNF-a) in plasma compared to patients with mild COVID-19 [45].

A key role in CRS is given to IL-6, which is synthesized by monocytes and macrophages in response to stimulation of TLRs. IL-6 is a pro-inflammatory regulator of T-cell functions, which stimulates the expansion of Th17 T-helper cells, inhibits the activation of regulatory T-cells (Treg) and promotes the development of a pro-inflammatory response [46].

Apart from the immune system dysfunction, hemostatic system disorders are of great concern for the COVID-19 pathogenesis. Examination of 183 patients with severe COVID-19 revealed hypercoagulation, accompanied by the development of thromboembolic conditions, which led to the occurrence of multiple organ failure. Most patients with a fatal outcome manifested an increased concentration of D-dimer and fibrin degradation products (FDPs), increased prothrombin time (PT) and activated partial thromboplastin time (aPTT), which was indicative of the disseminated intravascular coagulation (DIC) [47]. Administration of low molecular heparins in patients with severe COVID-19 contributed to the normalization of the D-dimer, PT and aPTT levels, and the decrease in the number of deaths, which was associated with the elimination of the DIC [48,49].

### 2.2. Prospects for Pharmacotherapy SARS-CoV-2

The targets for pharmacotherapy of SARS-CoV-2 may be its non-structural proteins (nsps)—3-chymotrypsin-like protease, papain-like protease and RNA-dependent RNA polymerase, which have high homology with proteins of other coronaviruses. Moreover, in the context of SARS-CoV-2 therapy, special attention is given to the ACE2 and TMPRSS2 proteins, as the main mediators of viral penetration into the cell. The regulation of the immune response, in particular, the production of cytokines to reduce the effects of CRS, also seems to be a promising method for pharmacotherapy of SARS-CoV-2. Drugs developed and used previously to treat patients with SARS and MERS are also considered for the treatment of COVID-19.

#### 2.2.1. Antimalarial Drugs

During the SARS and MERS epidemics, a common treatment strategy was to use chloroquine and hydroxychloroquine, used to treat malaria and systemic inflammatory diseases. Structurally, these drugs are derivatives of 4-aminoquinoline and have immunosuppressive and anti-inflammatory effects. Chloroquine and hydroxychloroquine are believed to block the penetration of the virus into cells by inhibiting glycosylation of the host receptors, proteolytic processing and endosomal acidification. However, there is no evidence to support the efficacy of chloroquine/hydroxychloroquine treatment of SARS or MERS. There are data proving that a hydroxychloroquine–azithromycin combination is effective against COVID-19. However, the number of patients with COVID-19 in this study was small: six patients with asymptomatic disease, 22 patients with upper respiratory tract infection and eight patients with lower respiratory tract infection. The authors of the study believe that hydroxychloroquine in combination with azithromycin contributed to the recovery of patients with SARS-CoV-2 on the 6th day of therapy in 60% of cases. It should be noted that azithromycin had the main therapeutic effect [50]. This result could not be reproduced in other studies [51,52]. According to various scientific groups, hydroxychloroquine was effective [53] or not effective [54] in the treatment of SARS-CoV-2 infection. All these works are distinguished by small numbers of patients and controversial criteria for evaluating the effectiveness of the results.

The largest retrospective study on the use of antimalarial drugs for COVID-19, conducted in the United States, involved 368 patients with COVID-19. Its effectiveness was assessed in terms of reducing mortality or preventing the deterioration of the patients. The average age of the patients was 70 years old; all of them were male. In addition to standard therapeutic measures, 97 patients received hydroxychloroquine; 113 patients received hydroxychloroquine in combination with azithromycin; 158 people made up the control group; their treatment was carried out without hydroxychloroquine. The need to use mechanical ventilation as a result of deterioration of the patient′s condition or death was chosen as the criteria for assessing the patient′s condition. In summary, 13.3% of patients who were prescribed hydroxychloroquine, 6.9% of patients who received it in combination with an antibiotic and 14.1% of control patients required mechanical ventilation; 11.4% of the control group patients died; among those who took hydroxychloroquine or its combination with azithromycin, death was registered in 27.8% and 22% of patients, respectively. A lethal analysis showed that hydroxychloroquine increased mortality in patients with COVID-19 [55]. In Russia, chloroquine and hydroxychloroquine are currently recommended for the treatment of COVID-19. However, the WHO did not issue recommendations for treatment with these drugs, suggesting only to report the results if used. 

#### 2.2.2. RAS Inhibitors

Patients with COVID-19 and concurrent diseases (cardiovascular disease, diabetes mellitus) are offered treatment with RAS blockers: ACE inhibitors (ACEIs) or angiotensin receptor blockers (ARBs). Patients treated with ACEIs or ARBs had increased ACE2 expression [56]. It is believed that the binding of SARS-CoV-2 to ACE2 can reduce the residual activity of ACE2, disrupting the balance of ACE/ACE2 towards increased ACE activity. This leads to pulmonary vasoconstriction and inflammatory and oxidative organ damage, and this, in turn, increases the risk of acute lung damage and systemic inflammation [36]. In this case, ACEIs and ARBs will help restore ACE/ACE2 balance. However, the use of ACEIs or ARBs can increase the binding of SARS-CoV-2 virus to ACE2 and stimulate disease progression [57].

It has been suggested that increasing the level of the soluble form of ACE2 in the blood can act as a competitive SARS-CoV-2 interceptor and inhibit the penetration of the virus into alveolar epithelial cells, protecting them from damage [58].

A clinical trial of Phase 2 administration of losartan (specific angiotensin II receptor antagonist of type AT1) is currently underway in patients with COVID-19 in an outpatient setting (ClinicalTrials.gov ID: NCT04311177) and an inpatient settings (ClinicalTrials.gov ID: NCT04312009) [59]. However, additional clinical and prospective studies are needed to find out if using ACEIs and ARBs can reduce the incidence and mortality rate of COVID-19.

#### 2.2.3. Protease Inhibitors

Lopinavir and ritonavir, 3-chymotrypsin-like protease inhibitors (3CLpro) used in HIV therapy, are also considered as a possible treatment option for COVID-19. In the life cycle of SARS-CoV-2, 3CLpro plays an important role, being one of the two proteases necessary for the proteolytic processing of coronavirus, and therefore, expression of viral genes [60]. The efficacy of lopinavir/ritonavir has been reported in several clinical case descriptions [61,62] and small retrospective non-randomized trials [63].

In a study by Cao et al. (2020) patients with COVID-19 were randomly divided into two groups: those who received standard therapy (control group), and those who received standard therapy accompanied by a combination of lopinavir and ritanovir (experimental group). The trial involved 99 patients in the experimental and 100 in the control group. The results of the use of the combination of lopinavir and ritanovir did not significantly differ from standard treatment, since an improvement in the condition of patients or their discharge from the hospital was observed on the 16th day on average in both groups. The number of patients with viral RNA in the blood and mortality in both groups also did not differ. Administration of lopinavir and ritonavir was often accompanied by side effects from the gastrointestinal tract, which led to an early exclusion of 13% of patients from the experimental group [64].

Another hypothetical treatment for COVID-19 is the inhibition of the TMPRSS2 protease, which SARS-CoV-2 uses to prime when it enters the cell. TMPRSS2 inhibitor camostat mesylate partially blocked the penetration of SARS-CoV-2 into cells of the Caco-2 and Vero-TMPRSS2 lines. Complete inhibition of virus penetration into cells was observed with the addition of the substance E64d, which is a nonspecific inhibitor of cathepsins B, H and L [65]. It is known that lysosomal cathepsins are necessary for the penetration of SARS-CoV and MERS-CoV viruses, and probably SARS-CoV-2, into the target cell via endocytosis [66].

Another study on the role of cathepsins in the penetration of SARS-CoV-2 into target cells used HEK293 cells, expressing the recombinant human ACE2 protein, as the model object. The cells were treated with a nonspecific inhibitor of cathepsin B, H, L and calpain (substance E64d), or a specific cathepsin L inhibitor (SID 26681509), or a specific cathepsin B inhibitor (CA-074). Treatment of cells with an E64d inhibitor reduced the penetration of SARS-CoV-2 into the cell by 92.5%. These data confirm that one of the cathepsins or calpains is required for SARS-CoV-2 to enter the cell. The use of a specific cathepsin B inhibitor did not affect the penetration of the virus into the cell, while treatment with a cathepsin L inhibitor reduced the penetration of SARS-CoV-2 into cells by more than 76%. This indicates the function of cathepsin L in priming the spike protein SARS-CoV-2 in the lysosome to penetrate into HEK293/hACE2 cells [67].

Calpain, cathepsin and TMPRSS2 inhibitors can interfere with the binding of SARS-CoV-2 virus to target cells. Several clinical trials of camostat and mesylate, TMPRSS2 inhibitors (ClinicalTrials.gov IDs: NCT04353284, NCT04374019, NCT04321096); and combinations of camostat, mesylate and hydroxychloroquine (ClinicalTrials.gov ID: NCT043385052) are currently in progress.

#### 2.2.4. Inhibitors of RNA-Dependent RNA Polymerase

Ribavirin is an inhibitor of viral RNA-dependent RNA polymerase, which explains the interest towards this inhibitor as a potential pharmacological agent during epidemics caused by SARS and MERS viruses. However, when studying the effectiveness of ribavirin against SARS-CoV, high concentrations of the drug were required to inhibit viral replication. Moreover, ribavirin had dose-dependent hematological toxicity, which makes it impossible to use it in high doses with a high frequency [68].

#### 2.2.5. Antibodies and Immunomodulators

As mentioned above, CRS is considered to be the main factor in the development of acute respiratory distress syndrome in patients with COVID-19, often leading to death. Antibodies to the IL-6 receptor (IL-6R), and tocilizumab, appear to be promising tools for the regulation of CRS. The Food Drug Administration (FDA) has previously approved tocilizumab for the treatment of rheumatoid arthritis [69]. Tocilizumab is currently undergoing clinical trials with COVID-19 (ClinicalTrials.gov IDs: NCT04315480, NCT04370834, NCT04332913, NCT04335071).

According to Drug.com (an online pharmaceutical encyclopedia), intravenous or subcutaneous administration of tocilizumab increases the risk of bacterial and fungal infections, especially tuberculosis. Therefore, monitoring of patients for the development of infectious diseases is recommended when using tocilizumab. This side effect may limit the use of tocilizumab in COVID-19. In addition, intravenous administration of tocilizumab may result in hypertension [70].

Melatonin, which possesses immunomodulatory, anti-inflammatory and antioxidant properties, can have a protective effect in acute lung injury and respiratory distress syndrome. Melatonin is believed to inhibit viral-induced apoptosis of the lung cells, which can cause acute lung damage [71,72,73]. Administration of melatonin in cases of COVID-19 may lead to the stabilization of the emotional state of the patient and improve sleep quality, thereby contributing to a positive impact on the recovery dynamics [71,72,73].

#### 2.2.6. Prospects for the Study of Peptides in the Complex Therapy of COVID-19

Currently, there are data available on peptides which are capable of inhibiting various phases of the SARS-CoV-2 life cycle, increasing the immune cells’ activity and normalizing the bronchopulmonary system functions in COVID-19 (Table 1).

Non-structural proteins SARS-CoV nsp10 and nsp16 were discovered to form a complex, in which nsp10 induces the methyltransferase activity of nsp16 by stabilizing the SAM-binding pocket and expanding the RNA-binding groove. In the nsp10 protein, the aa65–107 domain is necessary for binding to nsp16, and the aa42–120 domain is associated with nsp16 stimulation. The peptides GGASCCLYCRCH and FGGASCCLYCRCHIDHPNPKGFCDLKGKY obtained from the aa65-107 nsp10 domain can inhibit the 2-O-methyltransferase activity of the SARS-CoV complex nsp16/10. These data prove the relevance of the development of peptide drugs against SARS-CoV [74].

Heptad repeats (HR1 and HR2) are highly conserved sequences, located in the glycoproteins of the viral envelopes [75]. Peptides obtained from the HR regions of some viruses have been shown to inhibit the penetration of these viruses into the cell [76,77]. Based on this model, cell penetration inhibitors were identified for some viruses [78,79,80,81].

Molecular modeling showed that the S2 subunit in the spike protein SARS-CoV contains the HR1 and HR2 regions [82]. Based on these data, 25 peptides potentially capable of inhibiting the SARS-CoV penetration into the cell were constructed; their effects on the penetration of SARS-CoV viruses into 293T cells were investigated. The addition of peptides HR1-1 (NGIGVTQNVLYENQKQIANQFNKAISQIQESLTTTSTA) and HR2-18 (IQKEIDRLNEVAKNLNESLIDLQELGK) at concentrations of 0.14 and 1.19 μM led to 75% inhibition of SARS-CoV penetration into cells. It is possible that HR1-1 and HR2-18 can serve as functional probes for disrupting the mechanism of SARS-CoV-2 fusion with the membrane of the target cell [83].

During the COVID-19 pandemic in China, an immunomodulatory pentapeptide thymopentin (Arg-Lys-Asp-Val-Tyr, RKDVY, TP5), which is the active center of the thymus hormone, thymopoietin, was used as one of the therapeutic agents [84,85]. The main function of thymopentin is to normalize immunological parameters in cases of tumor, immunodeficiency and autoimmune diseases [86,87]. In vitro experiments showed that TP5 affects the functions of T cells and monocytes, increasing the level of cGMP secondary mediator and activating intracellular signaling cascades [88]. The ability of TP5 to normalize the functions of the immune system in viral diseases has been established [89]. In the context of SARS-CoV-2, thymopentin is considered a 3CLpro inhibitor. This peptide has a high affinity for the active site of this protease according to molecular modelling [90]. In 2003, thymopentin was used to treat patients with chronic bronchitis and SARS-CoV [90].

In Russia medicinal immunomodulatory drugs thymalin (registration number LS-000267 dated 26 February 2010, Ministry of Health of the Russian Federation) and thymogen (EW dipeptide, registration number LS-002304 dated 13 September 2011, Ministry of Health of the Russian Federation) have been used for a long time in cases of various diseases associated with impaired immune system functions [91]. Indications for thymalin application in children and adults: acute and chronic viral and bacterial infections; infectious purulent and septic processes; impaired thymus function, impaired regenerative processes; immune system and blood formation suppression after chemotherapy or radiation therapy in cancer patients. Indications for the administration of Thymogen are as follows: prophylaxis and complex therapy of acute and chronic viral and bacterial diseases of the upper respiratory tract; prevention of immune function and clot formation, and regeneration process suppression in the post-traumatic and postoperative periods; complex therapy of acute and chronic infectious and inflammatory diseases accompanied by a decrease in immunity.

The use of thymalin reduces the incidence of acute respiratory infections in senior patients by 2.0–2.4 times [92]. Studies on the effect of thymalin on the human respiratory system during early embryogenesis revealed its ability to accumulate in the epithelial cells of the respiratory tract [93]. Later studies showed that the use of thymalin and thymogen for the treatment of patients with acute lung abscess led to a decrease in the systemic inflammatory response syndrome, a decrease in plasma fibrinolytic activity and discontinuation of hypercoagulability [94]. In accordance with these results, thymalin and thymogen appear to be promising agents for COVID-19 therapy.

A double-blind, randomized, placebo-controlled study on the efficacy of thymogen in elderly patients after solid tumor removal in the abdominal cavity and retroperitoneal space was performed. The preoperative use of thymogen restored the structural and functional parameters of immunity and led to a significant decrease in the number and spectrum of postoperative complications and a reduction in the postoperative period [101]. Thymalin and thymogen manifested geroprotective properties in animal experiments and clinical studies. Thymogen administration to rats led to a 10% increase in survival and a decrease in the number of tumors by a factor of 1.5 [102]. Thymalin reduced mortality in senior patients over a 6-year observation period [95]. Thus, thymalin and thymogen normalize the functions of the immune and respiratory systems, and in addition, have a geroprotective effect. Given that the risk group for COVID-19 consists mainly of older persons, thymalin and thymogen can be considered as drugs for the complex treatment of infection caused by SARS-CoV-2.

Experimental use of thymalin and thymogen and their administration in patients with pneumonia of various etiologies and viral infectious diseases was accompanied by a normalization of the coagulation process and fibrinolytic activity of the blood [96,97]. Combined with the immunoprotective properties of these drugs, this may contribute to the effectiveness of treatment for patients with COVID-19 [98,99].

Currently, the assessment of the effectiveness of thymalin in COVID-19 is underway in Russia. A case of successful use of thymalin in severe coronavirus infection has been described [100]. A 60-year-old patient with chronic diseases (type II diabetes mellitus, stage III hypertension, ischemic heart disease) was admitted to the hospital a week after the appearance of the first signs of an acute respiratory infection. Upon admission, the patient was diagnosed with acute respiratory failure, dry cough and temperature 38.5–39.0 °C. According to the PCR test, the SARS-CoV-2 virus was detected in a nasopharyngeal swab. Computed tomography revealed that 75% of the lung tissue was affected. Based on these data, the patient was diagnosed with COVID-19.

The patient was put on a lung ventilator. Upon admission to the intensive care unit, the following drug therapy was prescribed: hydroxychloroquine, 400 mg, two times on the first day, followed by 200 mg two times a day per os; ceftaroline fosamil (antibiotic), 600 mg, two times a day, intravenously; levofloxacin (antibiotic), 500 mg, two times a day, intravenously; calcium nadroparin, 0.6 mL, two times a day, subcutaneously; paracetomol (antipyretic); ambroxol (expectorant). No positive dynamics were observed by the end of the first day in the hospital. Starting from the second day, thymalin was added to the basic therapy—10 mg one time per day intramuscularly for 7 d.

After the course of thymalin, the patient showed improvements in the blood test parameters (Table 2), gas exchange and respiratory function and a decrease in body temperature. According to the blood test, after the thymalin addition to the complex therapy, the relief of the “cytokine storm” was observed. Increases in the numbers of lymphocytes (2 times) and eosinophils (7 times) were revealed. At the same time, the concentrations of C-reactive protein, IL-6 and D-dimer in the blood decreased by 8, 2.5 and 7.2 times, respectively. This allowed the patient to be transferred from the intensive care unit to the infectious diseases department. On the 22nd day after being administered to the hospital, the patient was discharged. It should be assumed that the given clinical observation confirms the prospects for further research of thymalin in the complex therapy of COVID-19.

There is a possibility that the mechanism of action of thymalin and thymogen is similar to the mechanism of action of thymopentin. In this case, thymalin and thymogen are potential 3CLPro protease inhibitors. To establish homology in the structures of thymalin, thymogen and thymopentin, molecular modeling is required.

Oral administration of EDG tripeptide (Chonluten) and AEDL tetrapeptide (Bronchogen) is effective for the treatment of bronchopulmonary pathology (chronic obstructive pulmonary disease, chronic bronchitis with an asthmatic component). Oral administration of the EDG tripeptide resulted in an increase of the physical performance index and normalization of the organism functional state when exposed to low oxygen partial pressure. The tripeptide EDG has enhanced the effectiveness of standard therapy in patients with chronic bronchitis with an asthmatic component. The stress-protective effect of the EDG tripeptide is associated with its ability to regulate the expression of the c-Fos gene, the heat shock protein gene HSP70, the genes encoding the enzymes of the antioxidant system, SOD, COX-2 and the tumor necrosis factor gene TNF-α [108]. Regulation of TNF-α expression may help reduce CRS in patients with COVID-19.

The AEDL tetrapeptide regulates gene expression and protein synthesis of functional activity (Ki67, Mcl-1, p53, CD79, NOS-3, MUC4, MUC5AC, SftpA1) and lung cell differentiation (Nkx2.1, SCGB1A1, SCGB3A2, FoxA1, FoxA2) [109,110]. Thus, the peptides EDG and AEDL are effective for bronchopulmonary pathology. Their application may be considered among the methods of adjuvant therapy with COVID-19.

The KE dipeptide (Vilon) has an immunomodulatory effect. This peptide is capable of selective binding to the TCGA sequence of double-stranded DNA and manifests anticarcinogenic, antioxidant and geroprotective properties. It was shown that the KE peptide regulates the expression of the EPS15, MCM10 homologue, Culline 5, APG5L, FUSED, ZNF01, FLJ12848 fis, ITPK1, SLC7A6, FLJ22439 fis, KIAA69799, FIA, KIAA690099, FLA2 peptides, FLJ10914, Gdap1, MSTP028, MLLT3 and PEPP2, and the synthesis of cytoskeleton proteins, cell proliferation and metabolism, which explains its high biological activity [103]. The tetrapeptide AEDG (Epitalon) stimulates the activity of the neuroimmunoendocrine system, normalizing the functions of the thymus and cardiovascular system in pathology and ageing through regulation of melatonin synthesis [104]. Due to its ability to regulate melatonin synthesis, the AEDG peptide can be considered as one of the possible agents in complex COVID-19 therapy. It was previously found that the AEDG peptide activated the expression of the telomerase gene and contributed to an increase in length of telomeres in human blood fibroblasts and lymphocytes [105,106]. Activation of gene expression was accompanied by an increase in the number of cell divisions by 42.5%. This result correlated with an increase in life expectancy in animals by 42.3% after administration of this peptide [107]. The data presented indicate the geroprotective effect of the AEDG peptide.

Peptides are an innovative treatment option for an infectious disease caused by SARS-CoV-2. Short synthetic peptides or polypeptide complexes isolated from animal organs and tissues have practically no side effects and are effective in low concentrations. Unlike antimalarial drugs, RAS inhibitors, RNA-dependent RNA polymerase and some engineered immunosuppressants (tocilizumab), peptide drugs have practically no contraindications and can be used in patients with multiple chronic diseases (cardiovascular and renal pathology, diabetes mellitus, metabolic syndrome, pathology of the broncho-pulmonary system), including patients of older age groups. According to global statistics, severe COVID-19 and high mortality are observed in the aforementioned groups of patients [111].

One of the approaches used for the development of such drugs suggests the isolation of short peptides from the active structures of the virus proteins and further investigation of their antiviral activities. This approach was used in the study of human immunodeficiency viruses, Ebola and mouse coronavirus. Several peptides which inhibit the SARS-CoV penetration into the cell have been discovered; however, their clinical studies have not been conducted.

Immunomodulatory peptides can be considered as one of the promising methods for the COVID-19 treatment. During the pandemic in China, the immunomodulating pentapeptide thymopentin, which is the active center of the thymus hormone thymopoietin, was successfully used for COVID-19 therapy. Thymalin (a drug based on the thymus polypeptide complex) and thymogen (dipeptide isolated from thymalin) have similar effects. Thymalin and thymogen activate immunity in cases of immunodeficiency, and viral and bacterial infections; normalize the functions of the immune system and haemostatic system during ageing; and can be considered for the treatment of viral infection caused by SARS-CoV-2.

Some peptide immunomodulators, such as KE, AEDG and EW, are able to bind to the promoter regions of genes encoding cytokines, regulating their production and balance in the body. In cases of COVID-19, restoration of the cytokine balance is the most effective treatment strategy, since the violation of cytokine homeostasis in most cases leads to acute respiratory distress syndrome and subsequent death. Moreover, short peptides EDG and AEDL have proved their effectiveness in bronchopulmonary diseases, which indicates the possibility of their further study for the creation of drugs, effective in the complex therapy of COVID-19.

By summarizing the data on the molecular and physiological aspects of the pathogenesis of viral infection caused by SARS-CoV-2 and the prospects of peptide therapy for this disease, the following conclusion can be made (Figure 2).

Coronavirus SARS-CoV-2 binds to ACE2, TMPRSS and CD147 proteins on the membranes of epithelial cells of the lungs, kidneys, retina, small intestine, endothelium and smooth muscle cells of blood vessels and hepatocytes. Reproduction of the virus in the cells leads to their massive apoptosis. First of all, apoptosis of lung alveoli cells is observed, since the virus primarily affects the respiratory tract. Uncontrolled cell apoptosis leads to the activation of the synthesis of cytokines (IL-6, IL-10, IL-2, INF, TNF, MIP-1, MCP-1, IP-1, GCSF), the emergence of a strong inflammatory reaction, hypercoagulation, respiratory distress syndrome, multiple organ failure and death. This process may be aggravated by the ability of the SARS-CoV-2 virus to interact with hemoglobin in red blood cells, which leads to tissue hypoxia.

#### 2.2.7. Prospects for the Development of a Peptide-Based Vaccine Against SARS-CoV-2

There exists an opinion that the creation of a vaccine against SARS-CoV-2 will contribute to the creation of herd immunity, and this is of no less importance than the development of drugs effective against COVID-19. A vaccine based on 33 peptide epitopes of cytotoxic T-lymphocytes, T-helpers and B-lymphocytes is proposed for creation. Molecular modeling methods predict that such a vaccine will maintain a given spatial structure, is non-toxic and does not cause allergic reactions. Creation of a vaccine based on fragments of polypeptide regions of immune cell receptors will allow for recognizing the virus, and preventing its penetration into host cells and COVID-19 development [112]. Another study suggests creating a vaccine based on peptide epitopes of T and B-lymphocytes with a length of 20–30 amino acid residues. The simplicity of chemical synthesis, lower immunogenicity and cost of such a vaccine are the advantages of using a shorter polypeptide chain [113]. A method for the vaccine development based on peptide epitopes of T-lymphocytes is also proposed. Analysis of the structures of the E, S, N and M proteins of coronaviruses showed that they can be recognized by epitopes of T cells, which bind to the molecules of the major histocompatibility complexes HLA-I and II. Thus, peptide epitopes will be able to identify the main proteins of SARS-CoV-2 and activate the immune response [114].

## 3. Current Opinion

Short peptides can prevent the development of the pathological process with COVID-19 in three ways: virus protein inhibitor peptides (HR1-1, HR2-18, peptides based on aa65-107 domain structure analysis of nsp10 protein) are believed to inhibit virus replication; immunomodulating peptides (thymalin, RKDVY, EW, KE, AEDG) contribute to the normalization of innate and adaptive immunity, the hemostatic system and the synthesis of cytokines, and have an anti-inflammatory effect, thereby preventing the development of distress syndrome and multiple organ failure; additionally, peptides-bronchoprotectors (EDG, AEDL) can be used to reduce apoptosis, and activate proliferation and differentiation of bronchial epithelial cells.

Thus, one of the safest and most effective classes of substances suitable for further research and application in the treatment of coronavirus infection consists of three groups of peptides: SARS-CoV-2 protein inhibitors, immunomodulators and bronchoprotectors with a physiological mechanism of action.

## Figures and Tables

**Figure 1 molecules-25-04389-f001:**
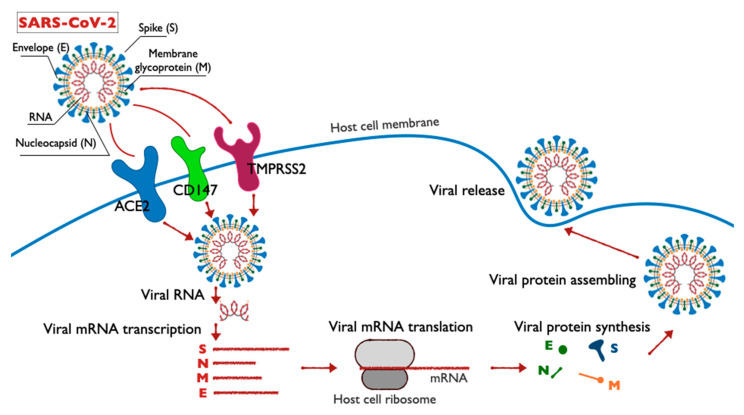
Schematic illustration of the structure, life cycle and molecular targets of SARS-CoV-2.

**Figure 2 molecules-25-04389-f002:**
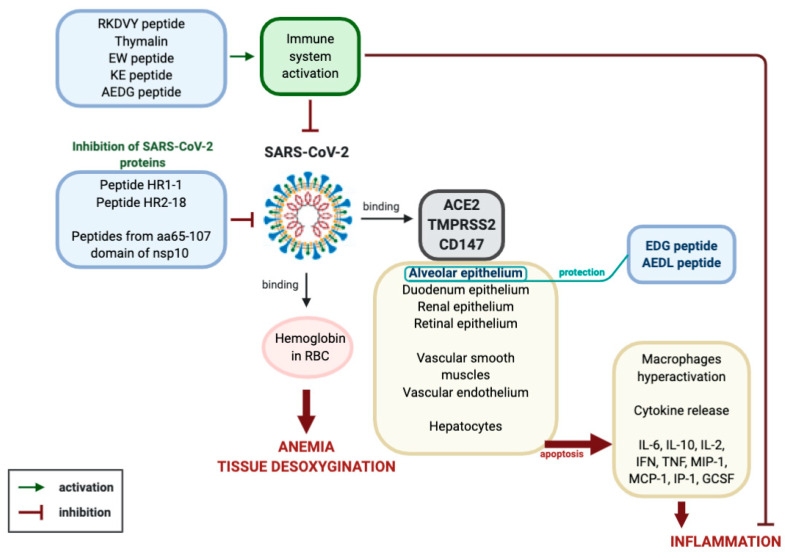
Expected mechanism of peptide regulation of pathological processes caused by the SARS-CoV-2 virus (explanation in the text).

**Table 1 molecules-25-04389-t001:** Compounds with potential biological activity against SARS-CoV-2 virus.

N	Peptide Name and Structure	Biological Activity
1	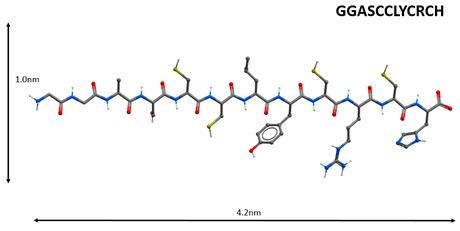	Inhibition of the enzyme activity of the virus 2-*O*-methyltransferase [74].
2	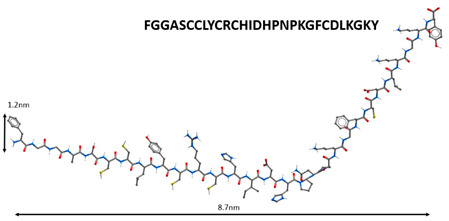	
3	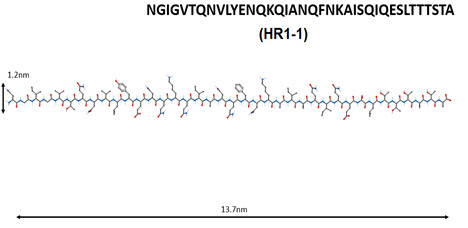	Inhibition of the virus binding to the host cell membrane [75,76,77,78,79,80,81,82,83].
4	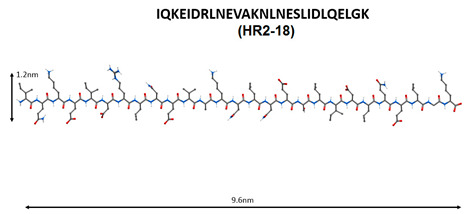	
5	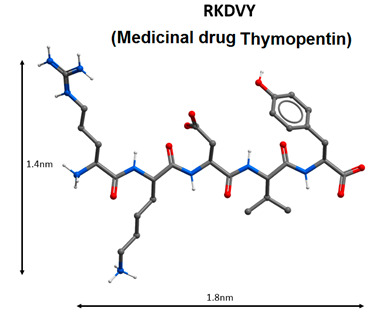	Activation of antiviral immunity, inhibition of the virus protein 3CLpro [84,85,86,87,88,89,90].
6	Polypeptide complex isolated from calf thymus (medicinal drug Thymalin)	Activation of antiviral immunity, bronchopulmonary system functions, hemostasis system. A case of successful clinical use in severe COVID-19 has been described [91,92,93,94,95,96,97,98,99,100].
7	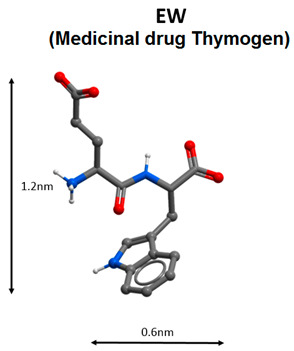	Antiviral immunity activation [91,96,97,98,99,101,102].
8	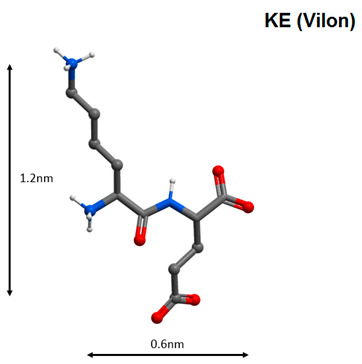	Activation of antiviral immunity and genes, which regulate the functional activity of the immune cells [103].
9	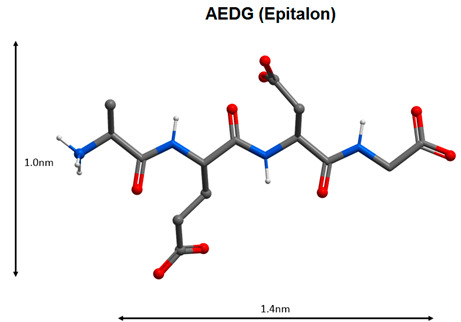	Activation of the neuroimmunoendocrine system functions, geroprotective effect [104,105,106,107].
10	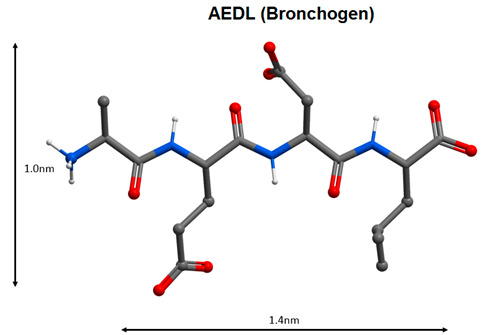	Regulation of the functions of the broncho-pulmonary and antioxidant systems [108,109,110].
11	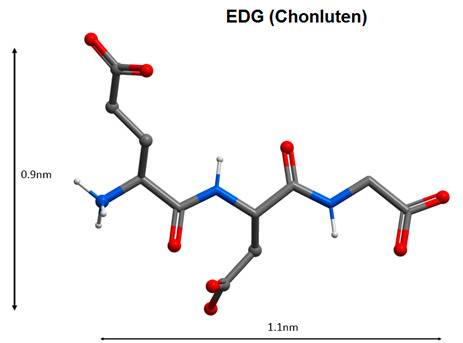	

Note: Nitrogen atoms are shown in blue, oxygen atoms in red, carbon atoms in gray, hydrogen atoms in white and phosphorus atoms in yellow.

**Table 2 molecules-25-04389-t002:** Analysis of a patient with COVID-19 before and after thymalin therapy (100 in modification).

Marker	Reference Range	Before Treatment	After a Course of Thymalin Therapy	Dynamic Pattern
Leucocytes, ×10^9^/L	4–9	4.37	7.58	+73%
Neutrophils, ×10^9^/L	2.0–5.5	3.8	5.93	+56%
Lymphocytes, ×10^9^/L	1.2–3	0.48	0.96	↑2-fold
Eosinophils, ×10^9^/L	0.02–0.3	0.01	0.07	↑7-fold
Platelets, ×10^9^/L	180–320	144	233	+62%
C-reactive protein, mg/L	0–5	48	6	↓8-fold
D-dimer, ng/mL	<243	2500	1000	↓2.5-fold
Fibrinogen, g/L	3.0	5.6	4.8	−17%
Activated partial thromboplastin time, s	24–34	53	38.6	−37%
IL-6, pg/mL	1.86–2.34	174	24.04	↓7.2-fold
CD3^+^, c/mL	880–2400	234	388	+66%
CD4^+^, c/mL	540–1460	145	239	+65%
CD8^+^, c/mL	210–1200	87	146	+67%
CD19^+^, c/mL	100–480	52	87	+67%
CD3^−^CD16^+^, c/mL	78–470	95	125	+32%

Note: ↑—increase, ↓—decrease.

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
