# Peer review of "Peptides: Prospects for Use in the Treatment of COVID-19"

_molecules, 2020, doi:10.3390/molecules25194389_

Round 1

Reviewer 1 Report

Manuscript entitled “Peptides: prospects for use in the treatment of COVID-19” discuss the worldwide problem, namely COVID-19 disease. Authors described pathological process related to infection and treatments used so far to battle this disease and suggest peptides can be studied and used in the treatment of coronavirus infection.

Authors provided comprehensive literature review on the COVID-19 disease but were unable to provide strong evidence that peptides can be used in the treatment of COVID-19 infection.

The section “2.4.4. Peptides in the treatment of COVID-19” only provide suggestions that discussed drugs …are promising therapies for COVID-19… or …can help increase the effectiveness of treatment… without information of their effectiveness in treatment of COVID-19.

Authors should rephrase their statement that this area needs further investigation.  

Author Response

Thank you for the important remark and the overall positive review of the article. We fully agree that peptides can only be considered as agents in the complex therapy of COVID-19 in the prospect, but this has not been proven so far.

The subtitle "Peptides in the treatment of COVID-19" has been changed to "Prospects for the study of peptides in the complex therapy of COVID-19"; the description of a clinical case featuring successful application of Thymalin in a patient with a severe form of COVID-19 has been added. The laboratory parameters of the blood test are described in the Table 2. The indicated additions are highlighted in blue.

More details please see the attachment.

Reviewer 2 Report

The manuscript is a review of the relatively current state of drug therapies for treatment of the severe form of Covid-19.  After a brief review of the molecular mechanism of the Covid-19 virus with its target receptors abundant in specific cells, the authors emphasize on the ineffectiveness of most of the current treatments with antimalarial drugs, some protein inhibitors, and inhibitors of the RNA polymerase of the virus, as well as a group of immunosuppressants. On the other hand, they stress on the potential use of 3 different groups of peptides as therapeutic agents (SARS-CoV-2 protein inhibitors, 533 immunomodulators, and bronchoprotectors), on the basis of their biological action and lack of side effects. Until submission date of this review manuscript, none of these peptides were specifically employed for the treatment of Covid-19 infection.

General Comments:

1) Regarding the lack of clinical and biochemical evidence, the effectiveness of the peptides as therapeutic agents for Covid-19 is highly speculative, subjective, overemphasized, and requires some moderation. The exaggerated emphasis can be only partly justified by the current state of the pandemic crisis.

2) This is an “Opinion Review”, rather than a review based on established research on the therapeutic effect of peptides on Covid-19 virus infection.

2)  There is definitely a potential for peptides (not only the peptide groups mentioned in the manuscript) as drugs against the Covid-19 virus, and the review is timely in this regard.

3) The manuscript reads monotonously, and suffers from lack of supporting figures on the molecular aspects of the mechanism of infection, immune system response, and representative molecular aspects of different drug functions (especially peptide drugs) mentioned in the text. I think a few more figures/tables on the molecular aspects of this review substantially improves and strengthens the content and makes it more accessible to the general audience.  

4) Since the subject of the manuscript is a rapidly developing area of research, some of the statistical information in the introduction section needs to be updated (e.g. the text between lines 58-62).

5) Any new developments in the use of peptides as therapeutic agents for Covid-19, need to be added to the relevant sections to make the manuscript less speculative.

6) Line 281 starting with “RAS inhibitors.” should be the heading of an independent subsection.

7) An overall language editing of the manuscript, especially in the introduction and conclusion sections, is recommended.

Author Response

Thank you for the important remarks and the overall positive review of the article.

The COVID-19 incidence data has been updated (September 2020), the English language of the article has been improved (highlighted in green), the typos have been eliminated. We agree that this is an "Opinion Review". To emphasize this, the “Conclusion” subheading has been changed to “Current Opinion”. 1 figure and 2 tables have been added to the article as recommended. Sections "Structure of SARS-CoV-2" and "Life cycle of SARS-CoV-2" have been deleted from the review. A diagram (Figure 1) illustrating these sections and the main molecular targets of SARS-CoV-2 in host cells has been added instead. To make the section on peptides more informative, Tables 1 and 2 have been added. The first table shows the spatial of the peptides described in the review and provides a brief information on their biological activity. The description of a clinical case featuring successful application of Thymalin in a patient with a severe form of COVID-19 has been added. The laboratory parameters of the blood test are described in the Table 2. The authors hope this will contribute to the overall informative and descriptive value of the article. Section “Prospects for the development of a peptide-based SARS-CoV-2 vaccine” has been added. In accordance with your remarks, “RAS inhibitors” has been moved out as a separate section. All the amendments and additions in the article are highlighted in blue.

More details please see the attachment.

Reviewer 3 Report

The manuscript entitled “Peptides: prospects for use in the treatment of COVID-19” by Vladimir et al. report current therapeutic molecules, including peptides for the SAR-CoV-2. The selection of topic for the review article is unique, and not many publications available in the literature. Therefore, this review article meets novelty for publication in the molecules. However, the manuscript lack several key features including more detailed discussion about peptides (figures, chemical structures, table) before it will be recommended for publication, as mentioned below.

  • Line 58: Data is from April 2020. It should be changed to the current month data.
  • Typos and grammars need to fix throughout manuscript.  
  • Line 63: There is no information for peptide. The text is mentioned for small molecules.
  • Line 32-355: There is no information discussed related to the peptide. Everything mentioned in these 7 pages are not helpful to the reader. The authors should consolidate it in a couple of figures/tables and write very concise information for other therapeutics and mechanism of SAR-CoV-2.
  • Line 356: 2.4.4. Peptides in the treatment of COVID-19. The author started discussing the peptide role in the manuscript. Figure 1 should come here and should be cited. Please elaborate on this section as it is what the manuscript should be. This is the place where the author has to use the schematic, the chemical structure of different peptides, figures, and tables. They also needs to include pharmacology, pharmaceutics, and pharmacokinetic properties of peptides mentioned in this section.
  • 456: Discussion about Melatonin is mixed with peptides. It should be separated.
  • 468: Conclusions. This title should change to another title. For example; current opinion or critical view.
  • Conclusion should be short (few sentences) and with the focus on peptides prospect.
  • Some vaccine information should be included as they were developed by using different peptide fragments. For example: (https://www.ncbi.nlm.nih.gov/pmc/articles/PMC7196559/, https://www.ncbi.nlm.nih.gov/pmc/articles/PMC7314509/, https://www.hindawi.com/journals/bmri/2020/2683286/, https://www.sciencedirect.com/science/article/pii/S0882401020305234)

Author Response

Thank you for the overall positive review of the article and important remarks, with which we fully agree.

The COVID-19 incidence data has been updated (September 2020), the English language of the article has been improved (highlighted in green), the typos have been eliminated. The purpose of the review has been reformulated, since, as justly noted, there had been no mention of peptides in the purpose of the review. Sections "Structure of SARS-CoV-2" and "Life cycle of SARS-CoV-2" have been removed. A diagram (Figure 1) illustrating these sections and the main molecular targets of SARS-CoV-2 in host cells has been added instead. The section "Prospective molecular mechanisms for the pathogenesis of coronavirus infection caused by SARS-CoV-2" has been significantly reduced: information on ACE2 protein gene polymorphisms and their relationship with the severity of COVID-19has been removed, as well as the information on the diagnostic role of the suPAR molecule in COVID-19. As justly noted, this information is not crucial for the review, but may hinder the perception of the material. To make the section on peptides more informative, Tables 1 and 2 have been added. The first table shows the spatial structures (constructed using ICM-Pro program) of the peptides described in the review and provides a brief information on their biological activity. Unfortunately, the authors could not find the data on the pharmacokinetics of the reviewed peptides – perhaps due to the fact that the study on these peptides has started recently. The exception is 3 peptide drugs (Thymopentin, Thymalin, Thymogen). The authors did not find the pharmacokinetics data for the Thymopentin pentapeptide and Thymogen dipeptide. Their application in clinical practice in relation to the review topic was added instead. Thymalin is a polypeptide extract (multicomponent drug); therefore, the pharmacokinetic study is not possible. Recently, it was discovered that Thymalin contains Thymogen and EW peptide, the properties of which were described in the review. Thymalin’s efficacy for COVID-19is currently being evaluated. This study has not yet been completed, but a case report for the use of thymalin in COVID-19 has been added to the article (Table 2). The description of melatonin as a possible protective agent in COVID-19 was moved from the section on peptides to the end of the section "Antibodies and immunomodulators". The subheading "Conclusion" was replaced by "Current opinion". A part of the summary information on the prospects for the study of peptides as potential agents in the complex therapy of COVID-19 was moved to the end of the "Prospects for the study of peptides in the complex therapy of COVID-19" section along with Figure 2 and a link to it. As per your recommendations, the “Current opinion” section now contains only a few phrases, which briefly summarize the content of the article. Section “Prospects for the development of a peptide-based SARS-CoV-2 vaccine”, which provides a brief analysis of the data from the articles referenced by the reviewer, has been added. All the amendments and additions in the article are highlighted in blue.

More details please see the attachment.

Round 2

Reviewer 2 Report

  1. The manuscript has been improved and reads better.
  2. In my version of the manuscript, Cyrillic alphabet and a different language (Russian) other than English is used in parts of pages 2 and 3. These and all other similar cases require translation into English.  

Author Response

We are very sorry for our mistake.

We corrected text in parts of page 2 and 3.

Reviewer 3 Report

Author revised manuscript significantly as per reviewers comments. However, few places Russian language were used. Also, the structure of peptides should show stereochemistry and side chain clearly than ball and stick model. Other than that I have no more question or revision. Thank you.

Author Response

Comment 1.  However, few places Russian language were used. 

Answer 1. We are very sorry for this mistake. We corrected text in page 2 and 3.

Comment 2. Also, the structure of peptides should show stereochemistry and side chain clearly than ball and stick model.

Answer 2. The structure of peptides shows in table 1 in 3D ball and stick model in ICM Program. This is the same, which you write. May be, you want to see peptides in the other view? If you want to see peptides in  the other view, would you be so kind to explane it more detail? We will be glad to do other view of peptide structure,if it needs. Thank you.